# An Investigation of Clock Skew Using a Wirelength-Aware Floorplanning Process in the Pre-Placement Stages of MSV Layouts

**B. Srinath [1], Rajesh Verma [2], Abdulwasa Bakr Barnawi [2], Ramkumar Raja [2], Mohammed Abdul Muqeet [2], Neeraj Kumar Shukla [2], A. Ananthi Christy [3], C. Bharatiraja [4,*] and Josiah Lange Munda [5]**

1   Consultant, MissionX Pvt, Mahindra 603002, India; srinath.bktr@gmail.com
2   Department of Electrical Engineering, College of Engineering, King Khalid University,
Abha, Asir 61411, Saudi Arabia; rkishor@kku.edu.sa (R.V.); abarnawi@kku.edu.sa (A.B.B.);
rmanoharan@kku.edu.sa (R.R.); mabdulmuqeet@kku.edu.sa (M.A.M.); nshukla@kku.edu.sa (N.K.S.)
3   Department of Electrical and Electronics Engineeqring, Saveetha School of Engineering,
SIMATS Saveetha University, Chennai 600077, India; chrisarun13@gmail.com
4   Department of Electrical and Electronics Engineering, SRM Institute of Science and Technology,
Chennai 603203, India
5   Department of Electrical Engineering, Tshwane University of Technology, Pretoria 0001, South Africa;
mundajl@tut.ac.za
*   Correspondence: bharatiraja@gmail.com

**Abstract:** Managing the timing constraints has become an important factor in the physical design of multiple supply voltage (MSV) integrated circuits (IC). Clock distribution and module scheduling are some of the conventional methods used to satisfy the timing constraints of a chip. In this paper, we propose a simulated annealing-based MSV floorplanning methodology for the design of ICs within the timing budget. Additionally, we propose a modified SKB tree representation for floorplanning the modules in the design. Our algorithm finds the optimal dimensions and position of the clocked modules in the design to reduce the wirelength and satisfy the timing constraints. The proposed algorithm is implemented in IWLS 2005 benchmark circuits and considers power, wirelength, and timing as the optimization parameters. Simulation results were obtained from the Cadence Innovus digital system taped-out at 45 nm. Our simulation results show that the proposed algorithm satisfies timing constraints through a 30.6% reduction in wirelength.

**Keywords:** timing constraint; multiple supply voltage; physical design; floorplanning

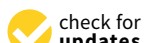



## 1. Introduction

The emergence of system-on-chip (SoC) technology has created remarkable impacts in mobile and wearable applications. This is mainly due to their high-speed processing and low power consumption. During the back-end design of these integrated circuits, the electronic design automation software (EDA) creates a clock network for the modules in the layout through routing from the clock source. The size of the clock network decides the time of the operation of modules which helps with high-speed operations. Since these ICs possess large numbers of modules, a compact packaging of modules with a reduced clock network size for a fixed die size introduces new complexities in the physical design process of SoC-based ICs and operates its modules with multiple supply voltages (MSV) for the reduction in power.

In MSV designs, the modules operating at the same voltage levels are floorplanned in a region called the voltage island for the reduction in the distribution of the power network area. Compared with single VDD designs, the aforementioned clock network size is pronounced in the MSV design. In spite of the conventional clock tree distribution techniques, a zero clock skew-based clocking methodology is necessary for the successful distribution of clock signals to the modules which contain the clock as one of their pins. The H tree, X tree, method of mean and median (MMM), recursive geometric matching

(RGM), and zero clock tree are some of the clock distribution methodologies available in EDA tools. These methodologies follow iterative methods for the distribution of clock signals from the source to the sink nodes in the layout.

*Related Works*

This section gives insights into works in literature which address and propose novel methods for simultaneous power reduction that satisfies the timing constraints. All the methodologies in these works were experimented on using the MCNC and GSRC benchmark circuits. Most of the previous research provides solutions to clock tree generation and distribution for a single VDD [1–3]. The implementation of methods for multiple supply voltage designs results in a large skew which degrades the speed of the operations performed in the IC.

Dynamic voltage frequency scaling (DVFS) and adaptive voltage scaling are the most effective techniques for power reduction, which function with a design that operates at different voltage modes [4]. Level converters are used in those designs to save power and improve speed [5]. To handle clocking strategies in the DVFS, the separate clock trees were generated for different operating modes. In order to reduce power consumption due to random logic circuits, a clustered voltage scaling scheme with row-by-row optimization in power was introduced [2]. In order to operate the IC within the predefined timing constraints, the critical mode optimization and surrogate-based optimization methods were proposed [3]. These methods insert buffers to meet the timing constraints in the clock path present in the voltage islands. In addition, this method reduces the setup violations before clock tree synthesis (CTS) since the buffers are inserted between modules without changing their positions. However, after detailed routing, it is observed that it increases wirelength; this will affect the performance of the chip. In order to overcome this trade-off problem, an algorithm named the deferred-merge embedding algorithm was proposed [6] which uses the DeFer algorithm for the optimization of wirelength.

Some of the works in the literature, such as [7–9], proposed methodologies for zero skew with sharp clock edge rates at the clock utilization points. Several design methodologies focused on techniques to optimize the process of clock tree synthesis [10–12]. Power gating [13], buffer sizing [14], and the insertion of multi-bit flip-flops (MBFFs) [15–17] were introduced for the reduction in power consumption and to satisfy the necessary timing constraints. In some designs, clock skew was also present in the intra levels of a clock tree. Adjustable delay buffers were inserted in interconnections to reduce the effect of this clock skew in the timing of the IC [18]. A legalization-based placement algorithm was also proposed [19,20] for an accurate timing analysis. In [15], MBFFs were used during the placement stage for the reduction in power and clock skew. In this method, modules in the layout were clustered for the reduction in clock skew in the clock distribution networks. Even though the clustering reduced the levels of the clock tree, it increased the intra-cluster delay. To resolve the routing complexities due to clock nets, the clock tree generated is segmented to achieve zero skew [21,22]. A power network distribution model was proposed in [23] for which the simultaneous optimized IR drop through power planning reduced wirelength. Since the clock distribution also had an effect on the latency of the design, a constraint-aware clock tree construction algorithm was proposed in [24–26].

In this paper, we propose a floorplan-aware clock tree generation methodology that identifies the clocked modules in the design and floorplans of those modules for balancing the clock tree. The modules of the floorplan are initially represented as a skewed binary tree (SKB) [27]. During the perturbation of modules, our proposed methodology considers ranking the clocked modules, which will reduce the length of the clock network.

The remainder of this paper report is organized as follows. Section 2 describes the SKB tree representation and its drawbacks. Sections 3 and 4 present an introduction to floorplanning representation and our proposed algorithm with its pseudo-code. Simulation and experimental results with IWLS benchmarks using the Cadence Innovus system are presented in Section 5.

## 2. Problem Formulation

Given a design of the initial floorplan consisting of its functional modules along with its operating voltage levels, we represent the modules in the voltage island using a skewed binary tree (SKB). Consider the floorplan of a simple design in Figure 1. This floorplan has modules which include the clock as a one of its pins. The re-positioning of these clocked modules in the floorplan results in geometric violations and increases the pre-defined width of the voltage island. This may lead to changes in the placement of modules in the neighboring voltage island that increases the routing resource in terms of wirelength [27,28]. For the purpose of quality floorplanning and to satisfy the voltage island constraint, we propose an algorithm to determine the optimal dimensions of the clocked modules in the voltage island. Then, we will incorporate the placement method analogous to the SKB tree for the positioning of modules in their respective voltage islands. To further improve the timing constraints, we will optimize the resulting floorplans iteratively for the reduction in the length of the clock tree. We will evaluate our resulting floorplan with a cost function which possesses wirelength, skew, delay, and power consumption as parameters. After the implementation of the proposed floorplanning methodology in the Cadence Innovus system, simulation results showed that our algorithm scales down the length of the clock tree through reducing the total wirelength in the design. As a result of repeated optimizations, our proposed algorithm also offers power saving and reduces delay compared to the existing SKB methodology.

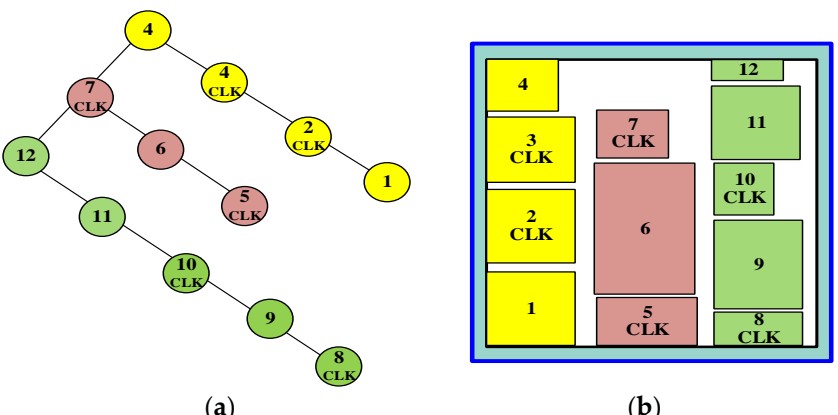

(**a**)                                 (**b**)

**Figure 1.** (**a**) Floorplan representation, and (**b**) placement of modules in the layout.

## 3. Preliminaries

Since our proposed methodology is based on SKB tree representation, first we review the floorplanning representation and the floorplanning that is used to satisfy the voltage island constraint in [29].

### 3.1. Floorplanning Representation

Given an initial floorplan with the module dimensions (width, height) and its operating voltage levels, we construct the SKB tree. For the floorplan shown in Figure 1a, we construct the SKB tree shown in Figure 1b. Each level of this tree represents voltage levels and then nodes in these branches are the modules operating at a respective voltage range.

### 3.2. Placement

The tree structure is traversed using the depth-first search process for the placement of modules from the left corner in the core area of the chip. In case of the formation of a voltage island, the width of the island is determined using the equation $W_i$.

$$W_i = \frac{a_i}{a_t} W_c (1 + \gamma) \tag{1}$$

In the above Equation (1), $a_i$ refers to the total area of modules in the power domain, $a_t$ refers to the total area of the chip, $W_c$ refers to the width of the chip, and $\gamma$ refers to the allowable dead space. A queue structure is maintained for the modules which fail to fit inside the estimated width of the voltage island. Before the placement of the next module, priority is given to the modules in the queue structure. Thus, the algorithm performs the placement process in the voltage island. The algorithm has a unique feature called a cluster constraint, to limit the density of modules in the voltage island. This feature helps to avoid a high density in the voltage island; thereby, it reduces congestion due to wirelength.

### 3.3. Perturbations

To satisfy the dead space constraint, the algorithm adopts the refinement of abnormal modules through rotation and perturbations. Three different perturbations were performed in this algorithm: the exchange of modules between voltage, inside the voltage island, and the change in the order of voltage islands.

### 3.4. Our Contributions

Our proposed algorithm undergoes the process explained in Section 4. Different from the SKB tree methodology, our proposed algorithm avoids cluster constraints so as to achieve less routing area. Additionally, our proposed algorithm avoids the refinement of modules in order to fulfill the current flow constraint during the routing stage.

## 4. Proposed Early Clock Planning Algorithm

In this section, we explain our proposed early clock planning algorithm for reducing the length of the clock tree.

Given an initial floor plan, we use SKB representation to arrange the modules of the floor plan. After the arrangement of modules, we identify the modules as having a clock pin. The algorithmic description of our proposed algorithm is given in the early clock planning algorithm (T).

The steps 1–2 involve inserting the modules in the tree structure as shown in Figure 1. The depth-first search process is used to traverse the tree which visits each module in the levels of tree T. Using the steps 3–6 for modules in every level of the tree, we use the function which determines the optimal dimensions of the clocked modules. In Step 7, if the search locates the clocked modules, it marks that module as *CLK* in the tree *T*. After distinguishing the modules with clock pins, we implement our proposed algorithm named $Clock\_Opt\_dimension\ (m_i, m_j)$ which identifies the suitable dimensions of the clocked modules and their relatively connected modules in the tree structure.

Using Algorithm 1 for all visited clocked modules, we identify the optimal dimensions. Since the module $m_{i-1}$ and $m_{i+1}$ may also connect with the clock pin module $m_i$, Algorithm 2 also finds the optimal dimensions of $m_{i-1}$ and $m_{i+1}$.

| **Algorithm 1:** Early clock planning algorithm (T) |
|---|
| 1.        A tree T, with nodes representing modules in the design |
| 2.        *level* $\leftarrow 1$, where *level* $\in T$ |
| 3.        for level do |
| 4.           $M \in T; m_i m_j \in M$ |
| 5.        Traverse tree T, using DFS; |
| 6.        mark clocked modules as CLK; |
| 7.        Choose $m_i$ and $m_j$ |
| 8.           $Clock\_Opt\_dimension\left(m_i, m_j\right)$; |
| 9.        update $m_i$ and $m_j$; |
| 10.       end for |

| **Algorithm 2:** *Clock_Opt_dimension* $\left( m_i, m_j \right)$ |
|---|
| 1.　　　　　for $m_i = CLK$ do |
| 2.　　　　　　　$Opt\_dimension(m_i, m_{i-1})$ |
| 3.　　　　　　　$Opt\_dimension(m_i, m_{i+1})$ |
| 4.　　　　　end for |
| 5.　　　　　Update $m_i, m_{i-1}$, and $m_{i+1}$ |

*Placement*

After obtaining the optimal dimensions of the clocked modules, it is updated in the tree structure so as to perform the placement of modules inside the core area of the chip as shown in Figure 2. Before the placement of modules, the width of the voltage island is obtained from the Equation (1). We undergo a similar placement process as in SKB after the implementation of our proposed methodology as it reduces the computational complexity.

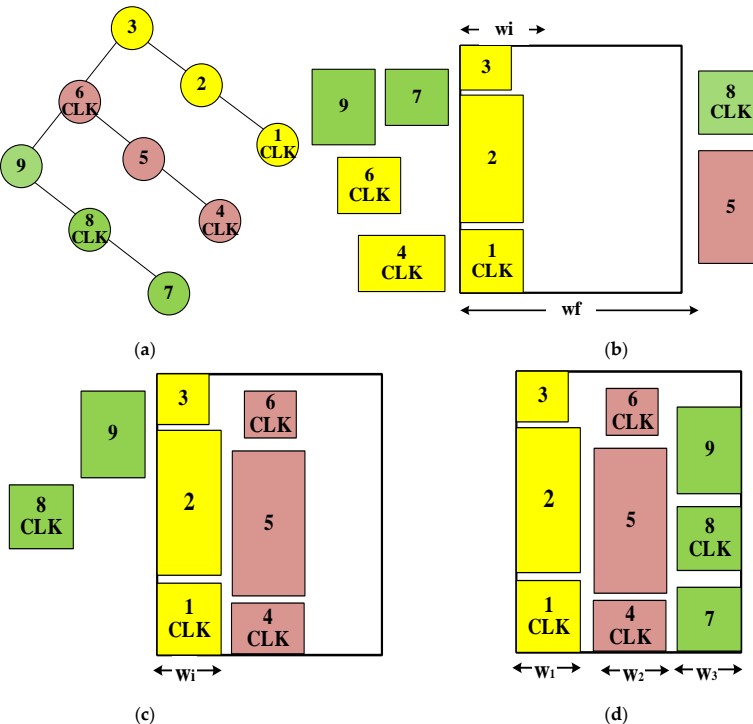

**Figure 2.** Placement process after implementation of proposed methodology. (**a**) tree structure; (**b**) placement of modules inside the core area of the chip 1; (**c**) placement of modules inside the core area of the chip 2; (**d**) placement of modules inside the core area of the chip 3.

Figure 2a shows the updated dimensions of modules in the tree structure. Figure 2b–d illustrates the arrangement of modules from the left corner of the chip. Even though the floorplan is with optimal dimensions, there is a need for optimization in order to reduce the dead space. For the reduction in this unused space, we optimize the resultant floorplan with simulated annealing with the cost function given in Equation (2).

$$Cost(F_1) = P + WL + D \tag{2}$$

To further improve the routing and avoid the skew induced due to routing, we optimize the floorplans with the cost function in Equation (3).

$$Cost(F_2) = P + WL + D \tag{3}$$

In the above Equations (2) and (3), $P$ refers to power in $nW$, $WL$ refers to wirelength μm, $D$ refers to delay in *pico* seconds, and skew is measured in *pico* seconds.

## 5. Simulation Results

In this section, we showcase the results of three different experiments which were performed on IWLS benchmark circuits. Table 1 shows the hardware description of IWLS benchmarks after synthesis in 45 nm technology.

**Table 1.** The hardware description of IWLS benchmarks for the Specification of design at the 45 nm node.

| Benchmark | Function | Sequential | Inverter | Buffer | Logic | Total |
|---|---|---|---|---|---|---|
| AES_CORE | AES Cipher | 530 | 5589 | 274 | 14,402 | 20,795 |
| DSP_CORE | 16-bit DSP | 3611 | 5258 | 42 | 23,523 | 32,436 |
| DMA_CORE | DMA | 2192 | 2678 | 253 | 13,995 | 19,118 |
| AC_97CTRL | WISHBONE | 2199 | 1525 | 111 | 8020 | 11,855 |

In this work, we use the size clock buffers of the tree to balance the generated clock tree. We use the engineering change order (ECO) in the Cadence Innovus EDA to choose the clock buffers. Currently, multiple supply voltage designs are the choice for the reduction in the dynamic power consumption of the high-speed ICs. Hence, in this work, we use low-threshold voltage standard cells from the technology library.

The experiment setup carried out using the Cadence Innovus is shown in Table 2. In the first experiment, we tend to perform iterations for the convergence of the cost function given in Equations (2) and (3) to enhance the performance of the proposed floorplan. In the second experiment, we compare our proposed floorplan with existing SKB tree methodology with three different aspect ratios to satisfy the fixed outline constraint. Finally, in the third experiment, we make a comparison of slack time, the number of sub-trees, and the number of levels in the clock tree with the existing method after clock tree synthesis (CTS).

**Table 2.** The hardware description of Cadence Innovus for the Specification of design at the 45 nm node.

| Benchmark Circuit | AES_CORE | DSP_CORE | DMA_CORE | AC_97ctrl |
|---|---|---|---|---|
| No. of Modules present | 17 | 28 | 15 | 16 |
| No. of Clocked Modules | 2 | 6 | 5 | 7 |
| Aspect Ratio | 1:1, 2:1, 3:1 | 1:1, 2:1, 3:1 | 1:01 | 1:01 |
| Core utilization | 70% | 70% | 70% | 70% |
| Supply Voltages(V) | 1.1, 0.9 | 1.1, 0.9 | 1.1, 0.9 | 1.1, 0.9 |

### 5.1. Performance of the Floorplan in Iterations

This study is further performed with various iterations of the floorplan based on proposed early clock planning methodology. We terminate an iteration if two consecutive cost function values are of the same value. Tables 3 and 4 show the floorplanning results for AES_CORE and DSP_CORE. The first column shows the number of iterations performed. Columns 2–4 show the results of total power consumption, delay, and wirelength in different power domains. Column 5 shows the cost function results which are calculated using the Equation (2). Figures 3 and 4 depict the layout after the implementation of the proposed methodology in AES_CORE and DSP_CORE.

**Table 3.** Iterative optimization in AES_CORE floorplan.

| Iterations | Total Power (nW) | | Delay (ps) | | Wirelength (um) | | Cost Function | |
|---|---|---|---|---|---|---|---|---|
| | PD-1 | PD-2 | PD-1 | PD-2 | PD-1 | PD-2 | PD-1 | PD-2 |
| Iteration-1 | 2.64 | 3.073 | 3.62 | 2.9 | 0.53 | 0.536 | 6.923 | 6.638 |
| Iteration-2 | 3.18 | 2.136 | 2.43 | 4.559 | 0.57 | 0.482 | 6.312 | 7.309 |
| Iteration-3 | 3.37 | 2.095 | 2.93 | 5.36 | 0.55 | 0.484 | 6.982 | 8.066 |
| Iteration-4 | 2.2 | 2.109 | 6.26 | 2.984 | 0.56 | 0.497 | 9.152 | 5.723 |
| Iteration-5 | 3.19 | 2.118 | 4.89 | 3.169 | 0.54 | 0.5 | 8.752 | 5.92 |
| Iteration-6 | 3.13 | 2.12 | 4.79 | 3.172 | 0.54 | 0.5 | 8.59 | 5.924 |

**Table 4.** Iterative optimization in DSP_CORE floorplan.

| Iterations | Total Power (nW) | | Delay (ps) | | Wirelength (um) | | Cost Function | |
|---|---|---|---|---|---|---|---|---|
| | PD-1 | PD-2 | PD-1 | PD-2 | PD-1 | PD-2 | PD-1 | PD-2 |
| Iteration-1 | 6.31 | 6.902 | 1.96 | 2.361 | 0.87 | 0.115 | 9.51 | 9.6 |
| Iteration-2 | 6.61 | 5.304 | 2.301 | 2.325 | 0.88 | 0.119 | 10 | 7.97 |
| Iteration-3 | 5.21 | 8.039 | 3.92 | 2.561 | 0.8 | 0.176 | 10.2 | 11 |
| Iteration-4 | 6.72 | 7.384 | 2.251 | 2.385 | 0.85 | 0.135 | 10 | 10 |
| Iteration-5 | 6.71 | 7.265 | 2.313 | 2.222 | 0.86 | 0.135 | 10.1 | 9.95 |

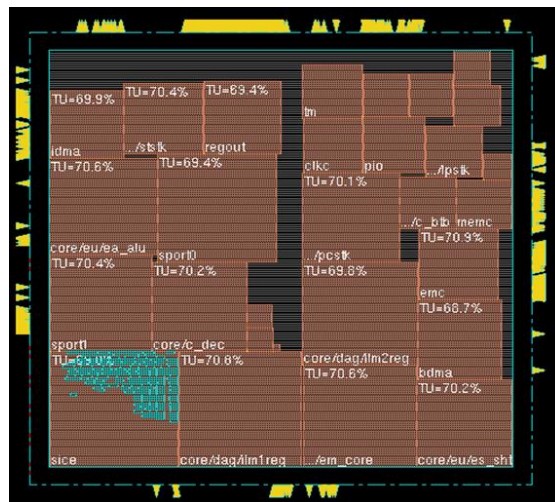

**Figure 3.** Layout: After implementation of proposed early clock floorplanning in AES_CORE.

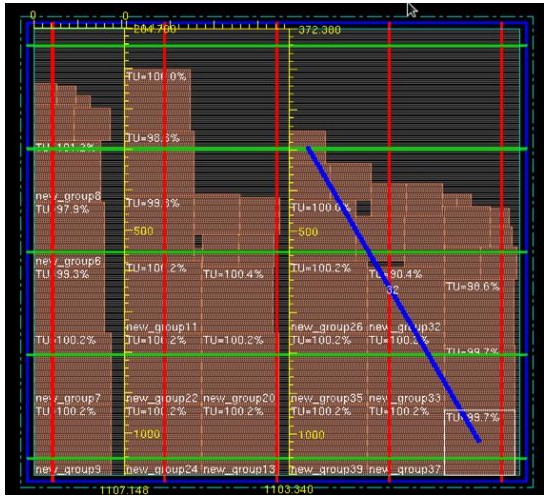

**Figure 4.** Layout: After implementation of proposed early clock floorplanning in DSP_CORE.

*5.2. Comparisons between the Proposed Floorplan Methodology and the SKB Tree-Based Floorplan after CTS*

Here, we compare our results after performing clock tree synthesis and calculate the number of levels, number of sub-trees, skew, and slack time of both the proposed and existing methods. Table 5 shows the skew results after the implementation of the proposed algorithm. For the reduction in induced clock skew, we optimize the resulting floorplans using the Equation (3).

**Table 5.** Skew results after CTS using our proposed methodology.

| Circuit | Levels | Sub-Trees | Skew | | Slack Time |
|---|---|---|---|---|---|
| | | | Rise Time (ps) | Fall Time (ps) | |
| 1-6 AES_CORE | 1 | 1 | 0 | 0 | 0.029 |
| DSP_CORE | 3 | 49 | 86.9 | 99.3 | 3.215 |
| DMA_CORE | 3 | 25 | 27.4 | 23.8 | 0.091 |
| AC97_ctrl | 5 | 53 | 18.2 | 18.1 | −1.3 |

Table 5 shows that the number of levels of the clock tree of the proposed method is less than the existing method in Table 6. We can observe that the number of sub-trees, skew, and slack time in the clock distribution depends on the number of levels of the clock tree. The lower the number of levels, the lower the sub-tree is, the lower the skew is, and the slack time will be close to zero. In Table 5 the columns 2–5 are our proposed method results and the columns 6–9 are the existing method. All the comparisons given in the table shows that our proposed method performs better than the existing method. Figure 5 shows the clock tree generated after implementing our proposed methodology in DSP_CORE.

**Table 6.** Skew results using existing methodology.

| Circuit | Levels | Sub-Trees | Skew | | Slack Time |
|---|---|---|---|---|---|
| | | | Rise Time (ps) | Fall Time (ps) | |
| 1-6 AES_CORE | 1 | 1 | 0 | 0 | 0.029 |
| DSP_CORE | 5 | 53 | 106.3 | 107.8 | −4.31 |
| DMA_CORE | 5 | 57 | 101.5 | 112.5 | −3.13 |
| AC97_ctrl | 7 | 64 | 28.2 | 29.3 | −2.89 |

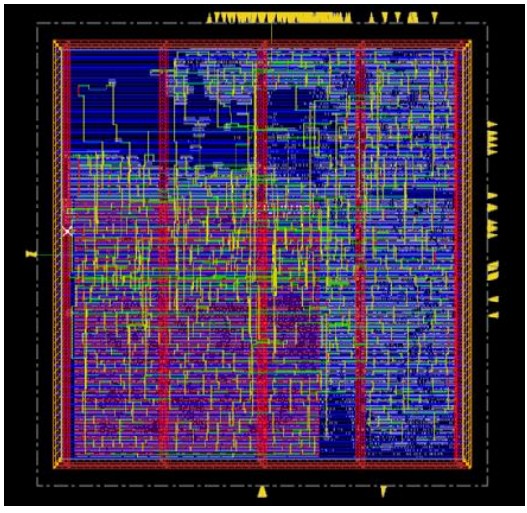

**Figure 5.** Layout: Clock tree after implementation of proposed methodology in DSP_Core.

## 6. Conclusions

The emerging VLSI integrated circuits and applications require a general methodology for the design of the chip for high speeds of operation and for processing the information. The hardware elements in the critical path of the design have impacts on power and timing constraints. In this paper, we focused on these two issues by presenting an early clock plan-based floorplanning algorithm that optimizes the fixed outline constraint, voltage island constraint, and reduces delay substantially. In the proposed framework with the

iterative optimization algorithm, it achieves practicality with the IWLS benchmark netlist synthesized at 45 nm. The experimental results reveal that the proposed floorplanning methodology shows significant improvements in clock skew, delay, and power saving though reduction in wirelength. 3D and monolithic-based integrated circuit designs are the most promising techniques for the compact fabrication of chips consisting of more than millions of transistors. Even though our proposed floorplanning methodology provides solutions to multiple supply voltage designs which aids in satisfying timing constraints by reducing the size of the clock tree, it needs to be reconstructed for 3D and monolithic ICs. This provides the motivation for our future work, which is to propose a novel physical design methodology which satisfies the desired timing constraints while designing 3D and monolithic ICs.

**Author Contributions:** Conceptualization, B.S., R.V., A.B.B., R.R., M.A.M., N.K.S., C.B. and J.L.M. and N.K.S.; methodology, B.S., R.V., A.B.B., R.R., M.A.M., N.K.S., A.A.C., C.B. and J.L.M.; software, B.S., A.A.C.,C.B., J.L.M.; validation, B.S., R.V., A.B.B., R.R., M.A.M., N.K.S., A.A.C., C.B. and J.L.M.; formal analysis, B.S., R.V., A.B.B., R.R., M.A.M., N.K.S., A.A.C., C.B. and J.L.M.; investigation B.S, R.V., A.B.B., R.R., M.A.M., N.K.S., A.A.C., C.B. and J.L.M.; resources, B.S., R.V., A.B.B., R.R., M.A.M., N.K.S.,C.B. and J.L.M.; data curation, B.S., C.B., J.L.M.; writing—original draft preparation, B.S. and C.B.; writing—review and editing, B.S., J.L.M. and C.B.; visualization, B.S., R.V., A.B.B., R.R., M.A.M., N.K.S., A.A.C., C.B. and J.L.M.; supervision, B.S., R.V., A.B.B., R.R., M.A.M., N.K.S., A.A.C., C.B. and J.L.M.; project administration, B.S., R.V., A.B.B., R.R., M.A.M., N.K.S., A.A.C., C.B. and J.L.M.; funding acquisition, J.L.M. and C.B. All authors have read and agreed to the published version of the manuscript.

**Funding:** The authors extend their appreciation to the Deanship of Scientific Research at King Khalid University, Kingdom of Saudi Arabia for funding this work through General Research Project under the grant number (RGP. 1/262/42).

**Data Availability Statement:** Experimental data is available upon request.

**Conflicts of Interest:** The authors declare no conflict of interest.

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
