# Peer review of "An Investigation of Clock Skew Using a Wirelength-Aware Floorplanning Process in the Pre-Placement Stages of MSV Layouts"

_electronics, doi:10.3390/electronics10222795_

Round 1

Reviewer 1 Report

The topic addressed Investigation of Clock skew using Wirelength-aware Floor- planning process in Pre-placement stages of MSV Layouts is potentially interesting, however, there are some issues that should be addressed by the authors:

The Introduction" sections can be made much more impressive by highlighting your contributions. The contribution of the study should be explained simply and clearly.

The authors should further enlarge the Introduction with current work about optimization algorithms to improve the research background, for example: Effective multi-sensor data fusion for chatter detection in milling process; Effective Feature Selection with Fuzzy Entropy and Similarity Classifier for Chatter Vibration Diagnosis; Optimal design of robust resilient automatic voltage regulators Towards Secured Online Monitoring for Digitalized GIS Against Cyber-Attacks Based on IoT and Machine Learning.

Clarify how you adjust the parameters of your planning algorithm

Increase the quality of figures 3-5

Conclusion section should be rearranged. According to the topic of the paper, the authors may propose some interesting problems as future work in the conclusion.

This study may be proposed for publication if it is addressed in the specified problems.

Author Response

  • Dear reviewer, thank you so much for your expert comments and suggestions to improve the quality of the manuscript. We have made all the requested changes and added the new information, as also given below in our response to your individual comments.

Note: The corrections in the revised manuscript are depicted in blue colour text

Reviewer 2 Report

Throughout the document, please edit to fix minor grammatical changes.

line 24: To provide context, indicate what is meant by timing constraint in this paragraph.

line 27: Indicate simulated annealing is used for optimization.

line 28: Spell out the “SKB” acronym. (This is first defined in lines 107-108)

line 31: “benchmarks circuits” appears to be grammatically incorrect in this sentence.

lines 120-121: Please provide the geographic location of the vendor (Cadence) and the version of the system used for this study.  Was Verilog or VHDL used for design entry?

line 156: Define “dead-space” in this context.

Equation (1): What is the design rationale associated with this equation?

Equations (2, 3): What are the constraints (max and min) associated with this cost function?

Figure 4: Please describe the lines on this image.

line 327: How were the skew results calculated?

Author Response

(The authors gave the same response as above.)

Round 2

Reviewer 1 Report

The authors should further enlarge the Introduction with current work about optimization algorithms to improve the research background, for example: Effective Nonlinear Model Predictive Control Scheme Tuned by Improved NN for Robotic Manipulators; Effective multi-sensor data
fusion for chatter detection in milling process; Effective Feature Selection with Fuzzy Entropy and Similarity Classifier for Chatter Vibration Diagnosis; Optimal design of robust resilient automatic voltage regulators Towards Secured Online Monitoring for Digitalized GIS Against Cyber-Attacks Based on IoT and Machine Learning. 

Increase the quality of figures 3-5